# Interfacial Imide Polymerization of Functionalized Filled Microcapsule Templates by the Pickering Emulsion Method for the Rapid Removal of 3,4,5-Trichlorophenol from Wastewater

**DOI:** 10.3390/nano12193439

**Published:** 2022-10-01

**Authors:** Zhuangxin Wei, Xinmin Ma, Pan Wang, Jianming Pan

**Affiliations:** School of Chemistry and Chemical Engineering, Jiangsu University, Zhenjiang 212013, China

**Keywords:** composite capsule adsorbent, Pickering emulsion, olive oil, 3,4,5-TCP

## Abstract

In this work, an olive oil-filled composite capsule (C–O/W) adsorbent was prepared for the adsorption of 3,4,5-trichlorophenol (3,4,5-TCP) by the emulsion templating method. Using methylene diisocyanate (HDI) and 1,6-hexanediamine (HMDA) as functional monomers, olive oil was encapsulated in a shell layer composed of graphene oxide and a polymer by interfacial imine polymerization. The contaminant target was efficiently removed by the hydrophobic interaction between olive oil and chlorophenols. The removal of 3,4,5-TCP was remarkable, with an encapsulation rate of 85%. The unique microcapsule structure further enhanced the kinetic performance, which reached 92% of the maximum value within 40 min. The adsorption of different chlorophenols was investigated using 2-chlorophenol (2-CP), 2,6-dichlorophenol (2,6-DCP), and 3,4,5-TCP. The adsorption of 3,4,5-TCP by the C-O/W microcapsules was found to be much higher than that of other chlorophenols. When analyzing a real sample, the content of 3,4,5-TCP was significantly reduced after adsorption by the C-O/W microcapsules, demonstrating that the C-O/W microcapsules were also capable of removing 3,4,5-TCP from a complex environment. This simple and inexpensive preparation strategy provides a new method for the synthesis of functionalized C-O/W microcapsule adsorbents and an effective adsorbent of 3,4,5-TCP.

## 1. Introduction

Chlorophenols are typical water pollutants that are widely present in wastewater solutions from drug, pesticide and dye production [1,2,3]. In China, a large amount of wastewater containing chlorophenols is discharged into natural rivers, causing serious pollution of surface water, groundwater and other water bodies. This has become one of the major water pollution problems in China [4,5]. Water pollution by chlorophenols is of great concern because of the acute toxicity, potential carcinogenicity, and moderate bioaccumulation of these chemicals, which can easily pose a serious threat to the human immune, respiratory, and reproductive systems [6,7,8]. Moreover, chlorophenols such as 2-CP, 2,4-DCP, and 3,4,5-TCP have been included in the priority list of pollutants for treatment by the US Environmental Protection Agency (EPA) and the European Community (EC) [9]. Adsorption is a separation technology for the efficient removal of organic and inorganic pollutants [10,11]. Inorganic adsorbents such as carbon-based, silica-based, and natural ores have a large specific surface area and microporous structure, as well as a good adsorption capacity for a wide range of organic pollutants [12,13,14]. However, the difficulty of their regeneration and their high cost limit their application. Therefore, there is an urgent need to develop some efficient and low-cost adsorbents for the effective adsorption of organic pollutants.

Olive oil is an inexpensive and green solvent that is widely used for the extraction and separation of water contaminants because of its hydrophobicity [15]. However, olive oil is a non-homogeneous and very complex mixture of compounds, its poor mobility makes it easy for water contaminants to be lost during the extraction process so that they usually remain in the water or in the solid residue left from the olive oil process, which makes the oil recovery difficult [16]. To solve this challenge, researchers have used encapsulation techniques to seal the olive oil in micro- and nano-capsules and improve its recovery and reutilization [17]. Common methods for capsule preparation include emulsion interfacial polymerization, interfacial crystallization, interfacial gelation and electroformation [18,19,20]. The construction of stable emulsion droplets is the first step to achieve micro-and nano-capsules. While Pickering emulsions use solid particles as stabilizers, the particle emulsifiers tend to deposit at the oil–water interface to reduce the interfacial tension and promote the formation of aqueous emulsion droplets [21]. They not only retain the basic properties of classical emulsions, but also have high chemical stability and good thermodynamic stability [22]. In addition, these emulsifiers help to limit the breakage of oil droplets and provide a strong physical barrier against their aggregation, providing them with long-term stability [23]. Therefore, Pickering emulsions offer advantages in dispersion compared to surfactant-stabilized emulsions, as they require relatively small amounts of particles to stabilize an emulsion, allow reducing the use of harmful surfactants, and are more environmentally friendly [24].

In this study, the stabilization of Pickering emulsions with graphene oxide, using methylene diisocyanate (HDI) and 1,6-hexanediamine (HMDA) as functional monomers, and the encapsulation of the green solvent olive oil in a shell layer composed of graphene oxide and polymer by interfacial imine polymerization were carried out and tested for the ability to remove chlorophenols from water. Specifically, graphene oxide (GO) nanosheets were used as Pickering particles, an olive oil solution containing methylene diisocyanate (HDI) as the inner phase and deionized water as the outer phase was prepared, and a stable O/W emulsion was obtained after emulsification; another monomer, 1,6-hexanediamine (HMDA), was then added to the outer phase to synthesize polyurea (PU) with a composite shell layer of GO to prepare an olive oil-filled composite capsule (C-O/W) adsorbent. Finally, kinetic, thermodynamic, adsorption equilibrium, adsorption regeneration and actual sample tests were performed on the prepared C-O/W adsorbent in the presence of 3,4,5-TCP to verify the ability of the olive oil-filled composite capsule (C-O/W) adsorbent to adsorb and separate chlorophenols from water.

## 2. Experimental Section

### 2.1. Materials

Graphite powder (GO), sulfuric acid (H_2_SO_4_), hydrogen peroxide (H_2_O_2_), sodium nitrate (NaNO_3_), sodium carbonate (NaCO_3_), nitric acid (HNO_3_), olive oil (C*_x_*H*_y_*O*_z_*), 1,6-hexanediamine (C_6_H_16_N_2_), hydrochloric acid (HCl), ammonia (NH_3_·H_2_O) were supplied by Sinopharm Chemical Reagents Company (Shanghai, China). Hexamethylene diisocyanate (C_8_H_12_N_2_O_2_), 2-chlorophenol (C_6_H_5_OCl), 2,4-dichlorophenol (C_6_H_4_OCl_2_), 2,6-dichlorophenol (C_6_H_4_OCl_2_), 3,4,5-trichlorophenol (C_6_H_3_OCl_3_) were purchased from Macklin (Shanghai, China). In addition, deionized water was used in all experiments, and the reagents were not purified before use.

### 2.2. Preparation of GO Nanosheets

GO nanoflakes were prepared using natural graphite powder as the main raw material by a modified Hummers method [25]. First, 1.0 g of graphite powder and 2.5 g of NaNO_3_ were slowly added to a three-necked flask containing 30 mL of concentrated H_2_SO_4_ and stirred for 30 min in an ice water bath (0 °C), and then 4.0 g of KMnO_4_ was slowly added and stirred for 2.0 h. Subsequently, the temperature was raised to 35 °C, and 40 mL of deionized water was slowly added; then, the temperature was raised to 98 °C, and the reaction was performed for 40 min until the solution was golden yellow. After completion of the reaction, 30% H_2_O_2_ was added to remove the excess KMNO_4_, after which the mixture was cooled with 140 mL of deionized water. Finally, the mixture was centrifuged, the supernatant and solid were separated, washed with 5% HCl several times and then with deionized water to neutral and finally freeze-dried to obtain the GO nanosheets.

### 2.3. Preparation of the C-O/W Microcapsule Adsorbent

Firstly, 10 mg of GO nanosheets described in 2.2 were dispersed in 5.0 mL of deionized water as the continuous phase, 208 μL of HDI was added to 1.0 mL of olive oil as the dispersed phase, and then 50 μL of 1.0 M sodium carbonate solution was added as the electrolyte. The preparation was emulsified by hand shaking for three cycles (hand shaking for 20 s with 15 s pauses), and then 0.5 mL of deionized water was added to dilute the formed emulsion. Subsequently, 193 mg of HMDA was dissolved in 1.023 mL of deionized water, added to the previously formed emulsion and left to stand for 5.0 h after three cycles of hand shaking (20 s hand shaking with 15 s pauses); the O/W emulsion was obtained, and the unreacted isocyanate functional group was quenched by adding a 5% dilute solution of ammonia and left to stand for 5.0 h. The C-IL/W microcapsules were prepared by centrifugation and vacuum drying.

### 2.4. Characterization

The pH of the solutions was measured by a pH meter (827 pH Lab Meter, San Francisco, CA, USA). The Fourier transform infrared (FT-IR) spectra of the samples were obtained by the Nicolet NEXUS-470 FT-IR apparatus (Los Angeles, CA, USA) and recorded using KBr pellets for solid samples. The morphology of microencapsulated sorbents was characterized using scanning electron microscopy (SEM, 6010PLUS/LA, Shanghai, China), and the emulsion droplets were measured by fluorescence microscopy (OLYMPUS-BX53M, Shanghai, China). A thermogravimetric analysis was carried out by a simultaneous thermal analyzer (STA449C, Wuxi, China) in conditions ranging from 100 to 800 °C with a speed of 10 °C/min under a nitrogen atmosphere. The concentrations of the phenolic compounds were determined by UV–vis spectroscopy (Cary 8454, Los Angeles, CA, USA) and high-performance liquid chromatograph (Agilent 1100, San Francisco, CA, USA).

### 2.5. Adsorption Isotherm and Kinetics Experiments

At 25 °C, 2-CP, 2,6-DCP, 2,4-DCP, 3,4,5-TCP test solutions were prepared at a concentration of 150 mg/L, and their pH was adjusted by 0.1 M HCl or 0.1 M NH4OH. Subsequently, 5 mg C-O/W microcapsules was added to 5 mL of the test solutions.

Adsorption isotherm experiments: Taking 3,4,5-TCP as an example, 5.0 mg of C-O/W adsorbent was added to 5.0 mL of 3,4,5-TCP sample solution (pH = 5.0), and the 3,4,5-TCP adsorption kinetics was tested by static adsorption at an initial concentration of 150 mg/L in the 3,4,5-TCP test solution. Then, the mixture was shaken in a constant-temperature water bath at room temperature, with a contact time of 0–120 min between the sample and the adsorbent. The adsorbent material was separated from the solution by standing, and the 3,4,5-TCP concentration in the filtrate was determined by UV spectrophotometry at 290 nm. The adsorption capacity *Q*_t_ (mg/g) at moment *t* was calculated by Equation (1) [26]:(1)$$Q_{t}=\frac{\left(C_{0}{\;-\;C}_{t}\right)V}{M}$$
where *M* (g) and *V* (L) represent the amount of adsorbent and the volume of the test solution, respectively, and *C_t_* (mg/L) is the concentration of 3,4,5-TCP in the filtrate at moment *t*.

Adsorption isotherm experiments: The adsorption equilibrium experiments were performed as described above for the kinetic experiments, with the initial concentration of the 3,4,5-TCP solution ranging from 10 mg/L to 500 mg/L. The mixture was then shaken and stirred in a constant-temperature water bath at room temperature for 1.0 h. The equilibrium adsorption capacity *Q*_e_ (mg/g) was calculated by Equation (2) [27]:(2)$$Q_{e}=\frac{\left(C_{0}{\;-\;C}_{e}\right)V}{M}$$
where *C_e_* (mg/L) is the concentration of 4,5-TCP in the filtrate at equilibrium.

In addition, a thermodynamic study was performed by adjusting the temperature of the water bath.

### 2.6. Cyclic and Actual Sample Adsorption Experiments 

Cyclic adsorption: Adsorption and desorption experiments can be used to assess the reusability of an adsorbent. In this study, the adsorption procedure was carried out according to the steps detailed above for the equilibrium experiments, and the adsorbent C–O/W after adsorption equilibrium was immersed in 15 mL of a deionized water/acetic acid mixture (9:1, *v*/*v*) for 48 h. Subsequently, the regenerated adsorbent was washed with distilled water to neutral pH and dried for the next adsorption–desorption cycle. The above steps were repeated 3 times, and the experiments were performed in parallel for 3 times.

Actual sample adsorption: Water samples were taken from Yudai River, Jiangsu University. The newly collected water samples were immediately filtered through a microporous nitrocellulose membrane (pore size of 50 nm) to remove suspended particles. The pH of the water samples was then adjusted to 5.0 by a 0.1 mol L^−1^ hydrochloric acid solution or a 0.1 mol L^−1^ ammonia solution. A spiked solution at a concentration of 50 μg/L was prepared. We added 15 mg of C-O/W adsorbent to 15 mL of the 3,4,5-TCP sample solution (pH = 5.0). The mixed solution was shaken for 1.0 h at 25 °C in a water bath and immediately filtered through a microporous nitrocellulose membrane (pore size 0.45 nm) to remove the suspended adsorbent. The filtrate was analyzed by high-performance liquid chromatography with an injection loop volume of 20 mL and a mobile phase consisting of deionized ultrapure water and methanol in a volume ratio of 40:60. The mobile phase flow rate was 1 mL min^−1^, and the oven temperature was 25 ℃. 

## 3. Results and Discussion

### 3.1. Design and Preparation of the C-O/W Microcapsules

The flow of the Pickering particles to construct olive oil-filled composite capsules is shown in Figure 1. In recent years, the problem of chlorophenol water pollution has been of great concern, and olive oil has been used for the extraction and separation of this chemical from water because of its green nature. Some researchers reported that the hydrophobic interaction between olive oil and chlorophenols has a great effect on the removal of chlorophenols. In this study, an olive oil solution with dissolved HDI was used as the oil phase, deionized water as the aqueous phase, and GO nanosheets as the stabilizing particles to form a stable O/W Pickering emulsion through emulsification by hand shaking. Subsequently, an aqueous solution of HMDA was added to the previously formed O/W emulsion in a certain proportion, and the HDI in the oil phase underwent in situ polymerization at the oil–water interface, rapidly synthesizing a strong polyurea shell layer (PU). The reaction is shown in Figure 1b. The polyurea obtained from diamine and diisocyanate polymerization at room temperature exists as an oligomer, but the shell layer combined with GO has a unique morphology, a high degree of polymerization, and strong water resistance. As a result, an olive oil-filled composite capsule encapsulated in a composite shell layer of graphite oxide and polyurea was prepared for the removal of chlorophenols from water. 

### 3.2. Characterization Analysis of the C-O/W Microcapsules

In order to identify the emulsion type, the oil-soluble dye Sudan III was added to the oil phase. After emulsification, the emulsion was observed under fluorescence microscope. Figure 2a shows the morphology of the emulsion in the bright field. Since graphene oxide has good hydrophilicity, it is partially dispersed in the aqueous phase and cannot be fully attached to the oil–water interface, resulting in uneven shape and size of the emulsion. In the dark field, the oil phase attached to the dye showed red fluorescence, which proved that the olive oil formed the internal phase, thus confirming the emulsion type as oil-in-water emulsion.

Figure 3 shows the SEM images of the GO nanosheets and C–O/W microcapsules. Figure 3a shows that the nanosheets were morphologically intact, and Figure 3 b shows a panoramic SEM image of the C–O/W microcapsules, which presented diverse morphologies and uneven sizes, with an average particle size of 90 μm, which could be caused by the fact that the hydrophilic graphene oxide nanosheets were mostly dispersed in the aqueous phase. Figure 3c shows a local magnification of the microcapsules; here, the graphene oxide and polyurea shell layers are clearly visible. In addition, the mechanical properties of the microcapsules were poor due to the uneven dispersion of the nanosheets. When observing the cross-section of the microcapsules, the shell layer appeared to be about 7 μm, which could be due to the polyurea shell layer formed by the in situ polymerization reaction of imine. These observaitons proved the successful preparation of C–O/W microcapsules.

We further investigated whether olive oil had been successfully encapsulated in the GO nanosheets and composite shell of PU. Figure 4 shows the FT-IR spectral results of the GO nanosheets and C–O/W microcapsules. For the GO nanosheets, the sharp peak appearing near 1729 cm^−1^ indicates the stretching vibration absorption peak of C=O; the characteristic peak of –OH near 3400 cm^−1^ proved the successful preparation of the GO nanosheets [28]. In contrast to the GO nanosheets, the characteristic peaks of C=O and -OH were present in the spectra of the C–O/W microcapsules, indicating that the GO nanosheets had been successfully attached to the microcapsules. The characteristic peak at 3300 cm^−1^ originated from N–H in PU [29]. In addition, the saturated C–H stretching vibration peak at 2925 cm^−1^ and the stretching vibration peak of C–O of triglycerides at 1165 cm^−1^ proved the presence of polyurea as well as of olive oil in the C–O/W microcapsules [30]. Based on the results of the above analysis, it was shown that the olive oil had been successfully encapsulated in the composite shell layer of GO nanosheets and PU. 

In order to investigate the thermal stability of the C–O/W microcapsules and the encapsulation rate of olive oil, the thermal degradation (TGA) behavior was evaluated in this study. In Figure 5, the weight loss curves of the C–O/W microcapsules and olive oil are basically the same, with decomposition starting at 380 °C, and thermal decomposition accelerating at 400 °C and finally remaining constant at 450 °C, which indicated a weight loss of olive oil encapsulated in the microcapsules [31]. The weight loss of 5.0 wt% in the C–O/W microcapsules from 200 to 380 °C was due to only the evaporation of water from the microcapsules, the decomposition of the PU shell layer, and the weight loss of GO functional groups [32]; 10 wt% of undecomposed residue remained after reaching 450 °C. Therefore, the encapsulation of olive oil in the C–O/W microcapsules reached a value of 85 wt%. The thermogravimetric results proved the successful preparation of the C–O/W microcapsules.

### 3.3. Effect of the Solution pH on the Adsorption of 3,4,5-TCP

The pH of a solution is one of the most important factors affecting the adsorption performance of an adsorbent, as it affects the solution chemistry of phenolic compounds, the activity of the adsorbent and the removal efficiency of CPs. Therefore, the adsorption of 3,4,5-TCP by the C-O/W microcapsules was evaluated by varying the pH of the test solution (2.0–7.0). Figure 6 shows the effect of the solution pH on the adsorption of 3,4,5-TCP. The amount of adsorption *Q*_e_ varied with increasing pH. In the pH range of 2.0–4.0, the effect of pH on the adsorption performance was small. An obvious change in the adsorption of 3,4,5-TCP by the C-O/W microcapsules occurred around pH = 5.0. This change may be related to the degree of ionization of chlorophenols in aqueous solution [33,34]. The pKa value of 3,4,5-TCP in aqueous solution is 7.20 at 298 K. Phenol can dissociate in solution and reach a dissociation equilibrium; when pH < 5.0, phenolic compounds exist mainly in molecular form. When pH > 5.0, phenolic compounds exist mainly in the form of negatively charged phenol ions, which leads to the disappearance of hydrogen bonds and thus to a decrease in adsorption. These experiments indicated that the adsorbent showed a good adsorption ability in the presence of the neutral molecular form of 3,4,5-TCP [35]. Therefore, the subsequent adsorption performance studies were carried out at pH = 5.0. 

### 3.4. Adsorption Kinetics Analysis 

The adsorption performance is inextricably linked to the adsorption kinetics and was assessed by determining the amount of the target contaminant adsorbed by the adsorbent for different contact times. The results are shown in Figure 7a. The adsorption capacity of the C-O/W microcapsules for 3,4,5-TCP increased rapidly from 0 to 20 min, which was attributed to the presence of a large number of adsorption sites in the adsorbent at the beginning of adsorption, and then reached 92% of the maximum capacity within 40 min. As the adsorption sites were gradually occupied, the adsorption capacity became saturated, and finally the adsorption equilibrium was reached after around 60 min. In order to gain insight into the adsorption kinetic process, quasi-primary and quasi-secondary kinetic equations were used in this study (Equations (3) and (4)), and the data were fitted and analyzed [36]: (3)$$Q_{t}{=Q}_{e}{\;-\;Q}_{e}e^{{-k}_{1}t}$$
(4)$$Q_{t}=\frac{k_{2}Q_{e}{}^{2}t}{{1+k}_{2}Q_{e}t}$$
where *Q*_e_ (mg/g) and *Q*_t_ (mg/g) correspond to the adsorption capacity at equilibrium and at time *t* (min), respectively. The values of the rate constants *k*_1_ (L/min) and *k*_2_ (g/(mg min)) were calculated from the fitted curves of ln(*Q*_e_ − *Q*_t)_ to *t* and *t*/*Q*_t_, respectively. Figure 7a shows the non-linear fits of quasi-primary and quasi-secondary kinetics of the C–O/W microcapsules for 3,4,5-TCP, and Table 1 summarizes the adsorption rate constants and regression values for both kinetic equations. Based on the quasi-secondary kinetic rate constants, the initial adsorption half-equilibrium time (*t*_1/2_, min) was calculated according to Equation (5) [37]: (5)$$t_{1/2\;}=\frac{1}{k_{2}Q_{e}}$$

Based on the fit of Figure 7a and the relevant data (*R*_2_) listed in Table 1, it was concluded that the actual adsorption process was more consistent with the quasi-secondary kinetic model, indicating that the adsorption process of 3,4,5-TCP by the C–O/W microcapsules relied mainly on chemical interactions. In addition, the actual equilibrium adsorption capacity *Q*_e,c_ (126.58 mg/g) calculated from the model fit was close to the experimental *Q*_e,e_ (125.81 mg/g) value, which also indicated that the quasi-secondary kinetic model could more accurately explain the adsorption process.

To further elucidate the adsorption mechanism, the adsorption–diffusion process was analyzed in this study. A linear fit to the data was performed according to the particle diffusion kinetic equation, using the fitted Equation (6) [38]:(6)$$Q_{t}=k_{\mathrm{id}}t^{1/2}+Ci\;$$
where *Q*_t_ (mg/g) is the adsorption capacity at time *t* (min), and *k*_id_ (mg·g^−1^·min^−0.5^) is the locus rate constant. 

### 3.5. Adsorption Isotherms and Thermodynamics Analysis

To further investigate the adsorption characteristics of the adsorbent in relation to the target, an adsorption equilibrium experiment was carried out in this study. Figure 8 shows the adsorption capacity of the adsorbent C–O/W microcapsules at different initial concentrations of the test solution when the adsorption equilibrium was reached. Subsequently, the experimental data for the adsorption of 3,4,5-TCP by the C-O/W microcapsules were fitted using the Langmuir and Freundlich adsorption isotherm equations. The nonlinear forms of the two equations are expressed in Equations (7) and (8) [39]: (7)$$Q_{e}=\frac{Q_{m}K_{L}C_{e}}{{1+K}_{L}C_{e}}$$
(8)$$Q_{e}{=K}_{F}C_{e}{}^{1/n}$$
where *Q_m_* (mg/g) denotes the maximum adsorption capacity, and *K*_L_ (L/mg) is the Langmuir adsorption constant. In addition, *K*_F_ (mg^1−n^·L^n^/g) is the adsorption constant in the Freundlich isotherm model, and 1/*n* is a constant for the adsorption conditions, indicating the measured values of support and surface inhomogeneity during adsorption, with values of 1/*n* < 1 reflecting favorable adsorption conditions.

Based on the curve fit in Figure 8 and the correlation coefficient (*R*^2^) in Table 2, the results of the Langmuir fit to the equilibrium data were found to be more consistent with the experimental results, indicating that the adsorbent surface had homogeneous active binding sites. According to the Langmuir calculations, the maximum adsorption capacities at 5 °C, 15 °C and 25 °C were 671 mg/g, 783 mg/g and 866 mg/g, respectively. In addition, the temperature plays a crucial role in the adsorption capacity of an adsorbent. This may be due to the fact that a high temperature facilitates the dissolution and diffusion of chlorophenol molecules, which leads to an increase in the adsorption capacity [40].

The adsorption thermodynamics can reflect the relationship between temperature and adsorption behavior and provide additional information on the adsorption of pollutants by adsorbent materials. Therefore, the adsorption process was analyzed in terms of enthalpy change (Δ*H*), entropy change (Δ*S*) and Gibbs function change (Δ*G*), using Equations (9) and (10) [41]:(9)$$\ln\;\left(\frac{Q_{e}}{C_{e}}\right)=\frac{\Delta S}{R}\;–\;\frac{\Delta H}{RT}$$
(10)ΔG =ΔH−TΔS
where *R* (8.315 (J/K·mol)) is the ideal gas constant, and *T* (K) is the temperature.

The linear curves fitted from the data are shown in Figure 9, and the relevant data calculations are presented in Table 3. The negative values of Δ*G* in the studied temperature range indicated that the adsorption of 3,4,5-TCP was a spontaneous process [42]. The positive value of Δ*H* indicated that the adsorption process in this study was heat-absorbing in nature, and therefore higher temperatures were favorable for the adsorption reaction. In addition, the Δ*S* values of the adsorption processes were all positive, indicating an increase in the stoichiometry of 3,4,5-TCP adsorption at the solid–solution interface. These results are consistent with those of the Langmuir isotherm. Thus, it was demonstrated that a high temperature is favorable for the removal of 3,4,5-TCP.

### 3.6. Adsorption Selectivity and Regeneration Performance Analysis 

In order to investigate the adsorption capacity of the C-O/W microcapsules with respect to different chlorophenols, 5.0 mg of adsorbent was placed into the test solutions of four chlorophenols (2-CP, 2,6-DCP, 2,4-DCP, 3,4,5-TCP) at the same concentration (150 mg L^−1^), and the supernatant was analyzed by UV spectrometry after 2.0 h of static adsorption at room temperature [43]. The amounts of 2-CP, 2,6-DCP, 2,4-DCP and 3,4,5-TCP adsorbed by the microcapsules were 27.55 mg/g, 56.12 mg/g, 78.25 mg/g and 123.26 mg/g, respectively, as shown in Figure 10. The results indicated an effect of the chloride substituent content in the chlorophenol. The higher the number of chlorine substituents, the more hydrophobic chlorophenols are, and according to the adsorption mechanism based on a hydrophobic interaction between olive oil and PEDs, the higher their adsorption by the adsorbent is. Since the C–O/W microcapsules showed the best adsorption ability towards 3,4,5-TCP, the subsequent experiments focused on the adsorption of 3,4,5-TCP.

The reusability of the adsorbent is an important indicator to assess its applicability. Therefore, in this study, three adsorption–desorption cycles were conducted with the C-O/W microcapsules as the adsorbent, adjusting the solution pH to 5.0. As shown in Figure 11, the adsorption capacity of the C-O/W microcapsules towards 3,4,5-TCP was 68 ± 1.0 mg/g after three cycles, which corresponded to about 53% of the first adsorption. Meanwhile, the desorption rates of the three desorption processes were 80%, 75% and 73%, respectively. The main factor affecting the adsorption process in this system could be the dispersion of the superhydrophilic GO nanosheets in the aqueous phase, which led to a decrease of the mechanical properties of the shell layer. After three cycles, it was still possible to maintain a high desorption rate. Therefore, it can be inferred that the proposed C-O/W adsorbent has a good regeneration performance and is a promising material for 3,4,5-TCP removal.

### 3.7. Actual Sample Analysis

The removal of 3,4,5-TCP from real samples containing various impurities is also an important criterion for evaluating the properties of adsorbents. In this study, water samples from the Yudai River at Jiangsu University were taken as the actual samples and used to prepare a 3,4,5-TCP solution at a concentration of 50 μg/L. A certain amount of adsorbent was introduced, and the mixture was subjected to a watershed shock for 1.0 h to promote the adsorption. The concentration of 3,4,5-TCP was subsequently measured using high-performance liquid chromatography. The results are shown in Figure 12. The original solution contained almost no 3,4,5-TCP, while the spiked solution showed a peak corresponding to 3,4,5-TC at about 4.0 min. After adsorption by the adsorbent, the peak decreased significantly. The above result indicated that the C-O/W microcapsules had the ability to remove 3,4,5-TCP also from an actual sample.

## 4. Conclusions

In this paper, olive oil-filled microcapsules were prepared as Pickering emulsion and interfacial in situ imine chemistry and used to remove chlorophenol contaminants from water. Firstly, the O/W emulsion containing the green solvent was prepared by using GO nanosheets as Pickering particles, olive oil as the inner phase and deionized water as the outer phase; subsequently, a highly polymerized polyurea shell layer was formed by interfacial imine chemistry of HMDA and HDI; finally, the contaminant target was efficiently removed by relying on the hydrophobic interaction between olive oil and chlorophenols. The C–O/W microcapsules with 85% encapsulation showed the best removal of 3,4,5-TCP in the adsorption process, reaching the adsorption equilibrium in 1.0 h. The maximum adsorption capacity of the microcapsules in relation to 3,4,5-TCP was calculated as 866 mg g^−1^ in the adsorption equilibrium experiment. Moreover, their adsorption capacity towards 3.4.5-TCP was demonstrated in real samples characterized by complex environments. This work provides a feasible strategy for the study of green solvent-filled microcapsules for the removal of CPs. 

## Figures and Tables

**Figure 1 nanomaterials-12-03439-f001:**
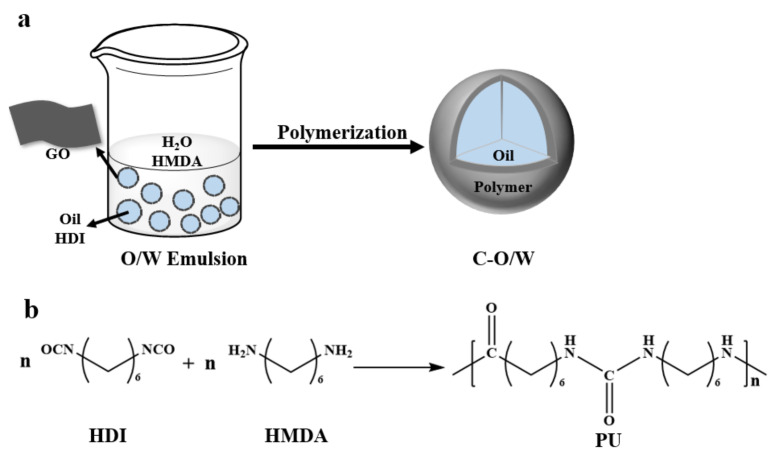
Schematic illustration of the synthesis of C–O/W microcapsules (**a**) and interfacial imine reaction in situ (**b**).

**Figure 2 nanomaterials-12-03439-f002:**
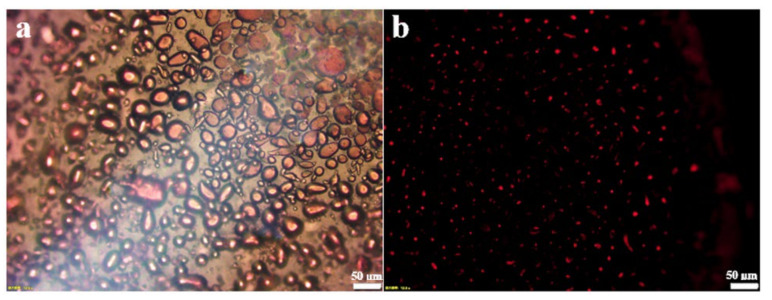
Fluorescence microscopy of the O/W emulsion in bright field (**a**) and dark field (**b**).

**Figure 3 nanomaterials-12-03439-f003:**
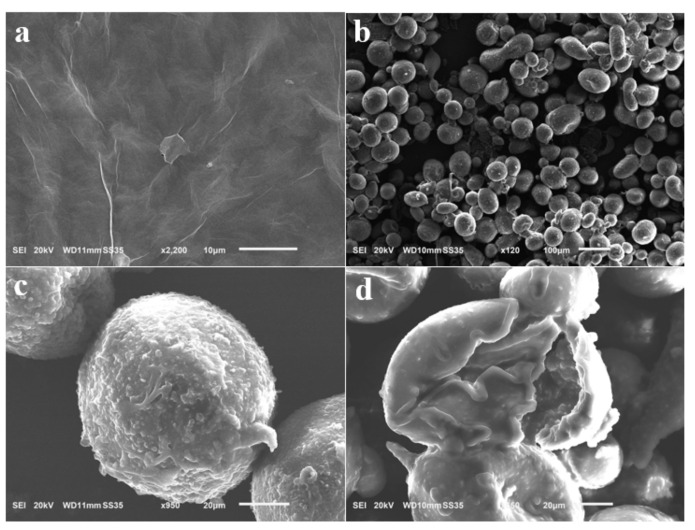
SEM image of GO (**a**) and C-O/W microcapsules (**b**–**d**).

**Figure 4 nanomaterials-12-03439-f004:**
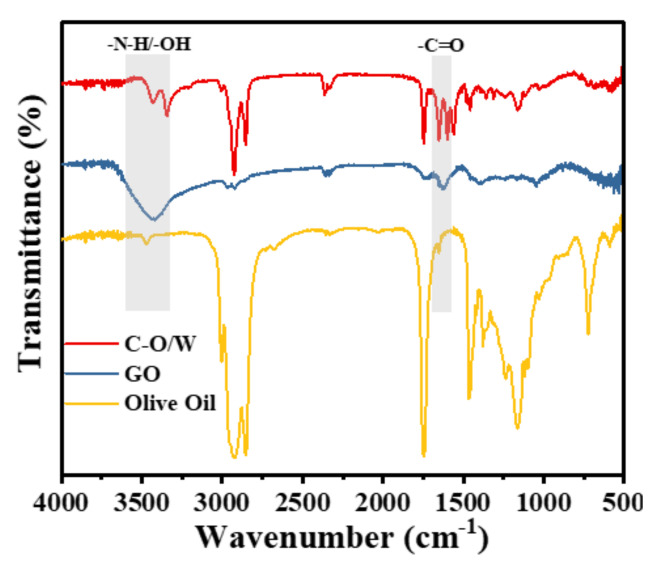
FT-IR spectra of GO, C-O/W microcapsules and olive oil.

**Figure 5 nanomaterials-12-03439-f005:**
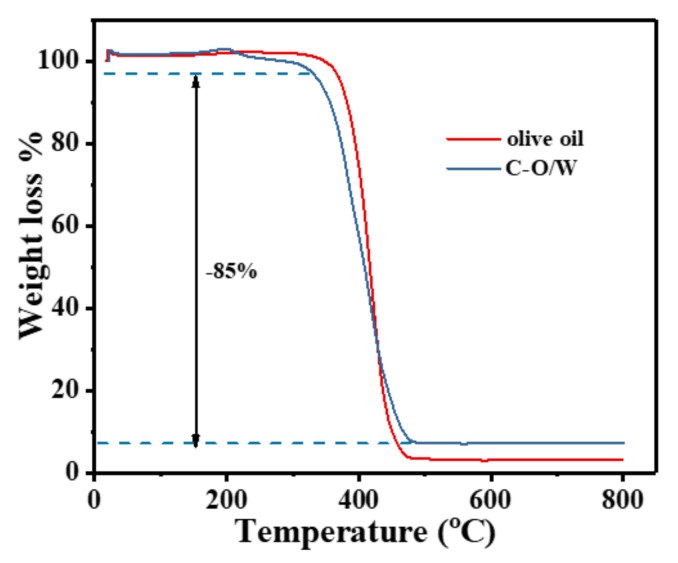
TGA curves of C–O/W microcapsules and olive oil.

**Figure 6 nanomaterials-12-03439-f006:**
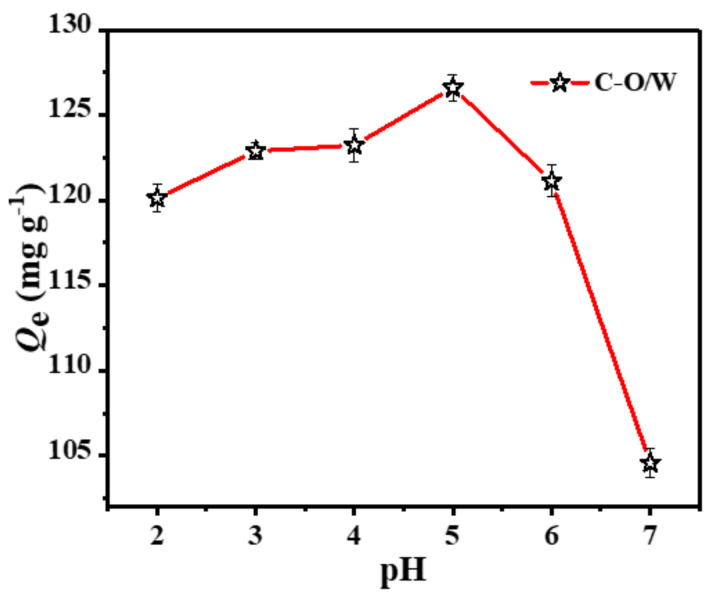
Effects of pH on the adsorption of 3,4,5-TCP by C-O/W microcapsules (*C*_0_ = 150 mg/L, T = 298 K).

**Figure 7 nanomaterials-12-03439-f007:**
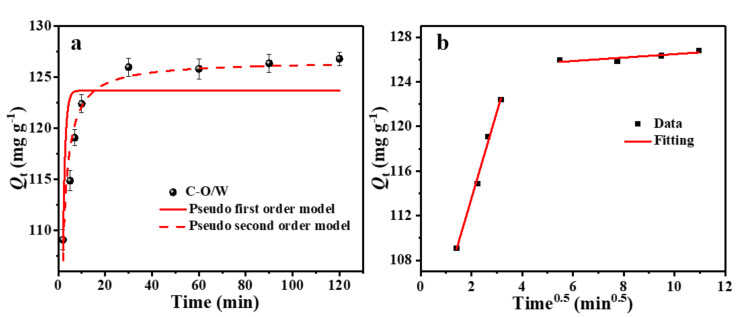
Non-linear kinetic model fitting curve for 3,4,5-TCP adsorption on C–O/W microcapsules (**a**) and particle diffusion (**b**) model fitting curves of 3,4,5-TCP adsorption by C–O/W microcapsules.

**Figure 8 nanomaterials-12-03439-f008:**
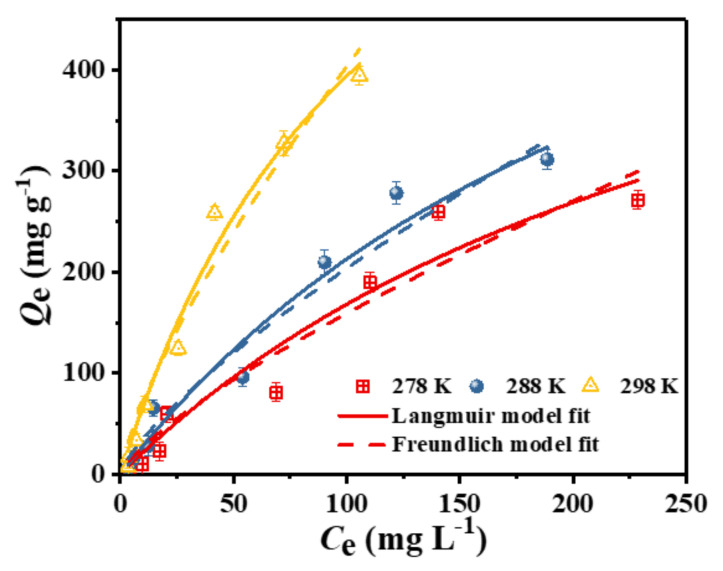
Adsorption isotherms of 3,4,5-TCP adsorption onto the C-O/W microcapsules and non-linear fitting curves according to the Langmuir and Freundlich equations (pH = 5.0, *T* = 278 K, 288 K, 298 K).

**Figure 9 nanomaterials-12-03439-f009:**
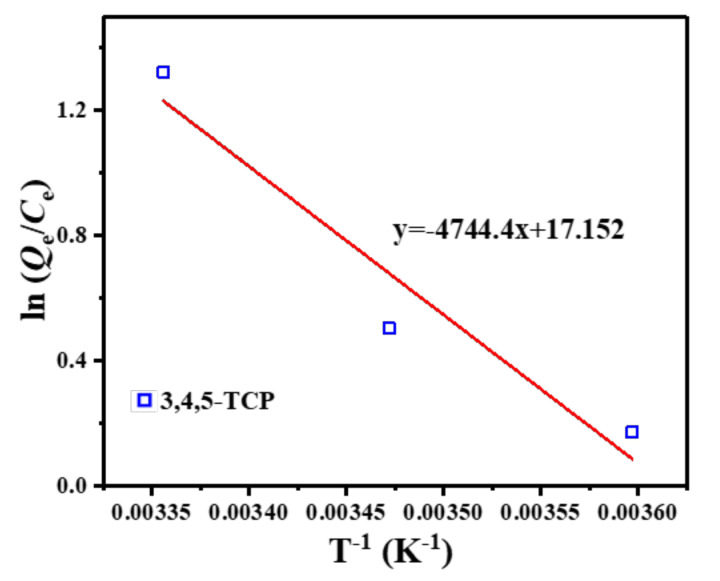
Van’t Hoff graph of 3,4,5-TCP adsorption onto C–O/W microcapsules.

**Figure 10 nanomaterials-12-03439-f010:**
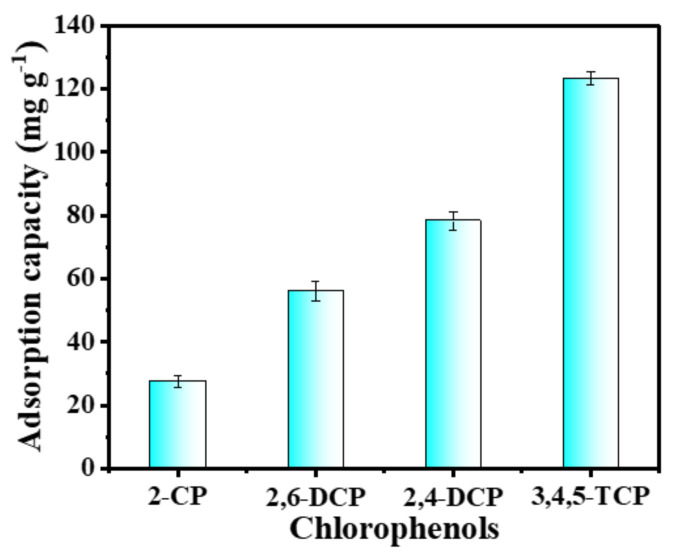
Adsorption capacity of the C–O/W microcapsules towards different PEDs.

**Figure 11 nanomaterials-12-03439-f011:**
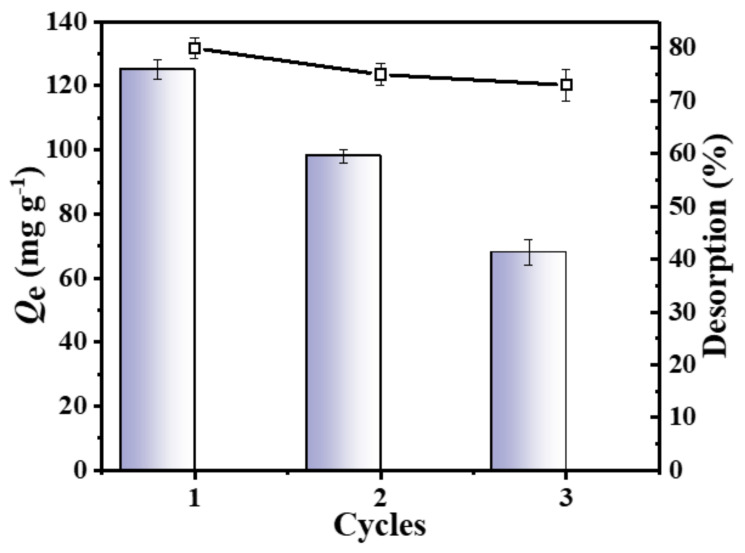
Adsorbed amounts of 3,4,5-TCP (down) and desorption efficiency (up) of the C–O/W microcapsules over 3 cycles.

**Figure 12 nanomaterials-12-03439-f012:**
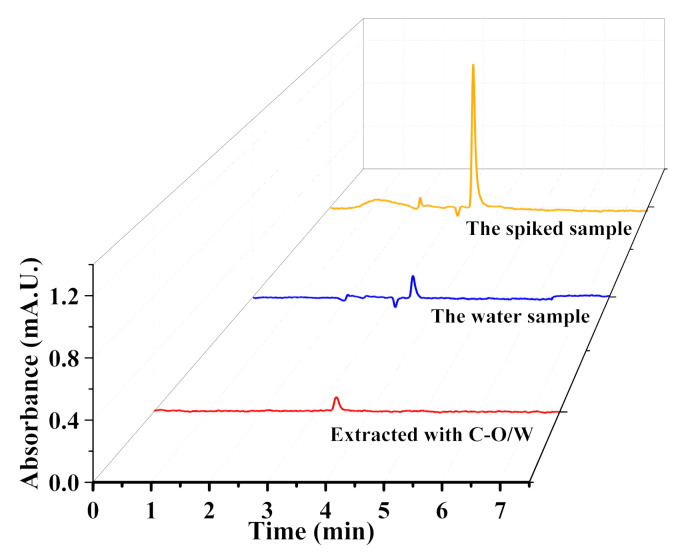
HPLC chromatograms of 3,4,5-TCP-spiked water samples.

**Table 1 nanomaterials-12-03439-t001:** Linear and nonlinear kinetic model parameters obtained in adsorption of 3,4,5-TCP onto C-O/W.

Kinetic Model				
Pseudo-first-order	*Q*_e,e_ (mg/g)	*Q*_e,c_ (mg/g)	*k*_1_ (min^−1^)	*R* ^2^
	125.81	123.67	1.034	0.5889
Pseudo-second-order	*Q*_e,c_ (mg/g)	*k*_2_ × 10^3^(g(mg·min))	*t^1/2^*	*R* ^2^
	126.58	0.58	13.88	0.9469
Particle diffusion equation	*k* _id1_	*k_id_* _2_	*R* _id1_ ^2^	*R* _id2_ ^2^
	7.742	0.156	0.992	0.921

**Table 2 nanomaterials-12-03439-t002:** Adsorption equilibrium constants for Langmuir and Freundlich isotherm equations.

Sorbents	*T* (K)	Langmuir Isotherm Equations			Freundlich Isotherm Equations		
		*Q*_m_ (mg/g)	*K*_L_ × 10^3^ (L/mg)	*R* ^2^	*K*_F_ (mg^1−n^·L^n^/g)	1/*n*	*R* ^2^
C-O/W	278	671	3.35	0.9294	4.700	0.7649	0.9267
	288	783	3.74	0.9587	6.075	0.7628	0.9606
	298	866	8.34	0.9700	12.70	0.7508	0.9561

**Table 3 nanomaterials-12-03439-t003:** Thermodynamic parameters for the adsorption of 3,4,5-TCP by C-O/W.

Adsorbed Target	*T* (K)	Δ*G* (kJ/mol)	Δ*H* (kJ/mol)	Δ*S* (J/(mol·K))
	278	−0.193		
3,4,5-TCP	288	−1.619	39.45	142.6
	298	−3.045		

## Data Availability

Not applicable.

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
