# Peer review of "Interfacial Imide Polymerization of Functionalized Filled Microcapsule Templates by the Pickering Emulsion Method for the Rapid Removal of 3,4,5-Trichlorophenol from Wastewater"

_nanomaterials, 2022, doi:10.3390/nano12193439_

Round 1
Reviewer 1 Report
The specific comments are given in the attached .pdf file.

Author Response
Responses to the Reviewers’ Comments
Dear Prof. Serena Song:
Thank you very much for your letter and for the reviewer’s comments concerning our manuscript entitled “Interfacial Imide Polymerization of Functionalized Filled Microcapsule Templates by Pickering Emulsion for Rapid Removal of 3,4,5-Trichlorophenol from Wastewater” (Manuscript ID: nanomaterials-1924018). We are truly grateful to yours and reviewers’ critical comments and valuable suggestions. Those comments are all valuable and very helpful for revising and improving our paper, as well as the important guiding significance to our researches. Based on these comments and suggestions, we have tried our best to improve the manuscript and made some changes in the revised manuscript in those days. And in the “Response to Decision Letter”, all the changes and also highlighted with red in the revised manuscript and the Supporting Information (corresponding to the serial number in the list). We appreciate for the Editors and Reviewers’ warm work earnestly, and hope that the correction will meet with your approval. Below you will find our point-by-point responses to the reviewers’ comments or questions.
We deeply appreciate your consideration of our manuscript, and if you have any questions, please don’t hesitate to contact us.
Thank you and best regards.
Yours sincerely,
Jianming Pan
Responses to the Reviewer 1’s Comments
The submitted paper concerns the Pickering emulsion-based capsules fabricated for efficient removal of chlorophenols from water. The Authors performed numbers of experiments to characterize their capsules and show an efficient adsorption of contaminants. However, the minor revision of the manuscript is required. The specific comments are given below.
Comment 1: In the Introduction, the Authors said that “the mobility of olive oil makes it easy to be lost and difficult to be recovered” (line 42). As it is important as a motivation of the research, this issue could be more elaborated.
Response 1: Thank you very much for the encouraging comments and constructive suggestions. We have addressed this issue in more detail, and all changes in the revised manuscript are highlighted with a red background.
Comment 2: When describing Pickering emulsions, the Authors suggested that particles attached to the interface reduce the surface tension (line 49). It is commonly known that this a mechanism of stabilization by using surface active agents, but how to explain this for particle-stabilized emulsions? Please, reconsider this issue and add more specific information if needed.
Response 2: Thanks for your suggestions. For the emulsion system, the function of the surfactant is to reduce the surface tension of the oil-water interface so as to achieve the effect of stabilizing the emulsion.
Comment 3: Figure 3 (it should be a full word for Figures in the text of manuscript according to the guidelines for the authors for this specific journal) presents SEM images of the capsules. When describing the results, the Authors mentioned about the shell thickness of 7 micrometers. What about the size of capsules? The additional characterization of capsules when it comes to their size is recommended. Moreover, it is quite interesting, how the size of capsules influences the efficiency of removing pollutants. The Authors should add such a comparison with the existing literature data if possible.
Response 3: Thinks for your suggestions. The work in this manuscript employs an interfacial imine polymerization method to encapsulate olive oil in a shell composed of graphene oxide and polymer. In terms of characterization, the efficiency of removing contaminants is almost exclusively related to shell thickness and has little to do with capsule size.
Comment 4: In Figure 5, the Authors presented the results from TGA. However, the description of the experimental setup seems to be missing in Materials and Methods section (line 107). What is more, it is not clear how this experiment looked like for capsules, for instance, whether the capsules were immersed in water as carrier fluid or dried. Are there some specific changes in capsule structure when exposed to high temperature? It is also important as thermal annealing is one of the methods for preparation of capsules from Pickering droplets, see, e.g., https://doi.org/10.1039/C7SM00528H and https://doi.org/10.1016/j.colsurfb.2020.111070.
Response 4: Thinks for your suggestions.
- A description of the TGA's experimental equipment has been added to the manuscript and highlighted in red.
- For the TGA experiments, the capsules were immersed in water as the carrier liquid.
- In this experiment, the capsule is mainly used for sewage treatment in nature and does not need to work in a high temperature environment. And exposed to high temperature, because the ionic liquid has high stability, the structure and size of the capsule will not change. This can refer to the following article on "The stability of polyurea ionic liquid":
- Wang, X. Ma, Q. Li. Green synthesis of polyureas from CO2 and diamines with a functional ionic liquid as the catalyst. RSC advances 2016, 6, 54013-54019.
Comment 5: In Figure 7, the Authors showed the fitting of the experimental data to the Eq. 3 and 4. It might be helpful for the readers to mention about these equation in the figure caption.
Response 5: Thinks for your suggestions. We have tried to put the formula in the legend, but found that this may affect the layout of the whole article, and most of the formula exists in the explanation of the manuscript, I hope you can understand.
Comment 6: When analyzing the results in Figure 9, the Authors concluded that “the high temperature is favorable for the removal of chlorophenols” (line 340). What is the expected performance of the capsules as adsorbents for much higher temperatures (higher than room temperature considered in Figure 9)? Is this a monotonously positive correlation?
Response 6: Thinks for your suggestions. Figure 9 demonstrates that high temperature is good for the removal of 3,4,5-tcp, and for your question, it is obviously more beneficial for 3,4,5-tcp removal at higher temperature, but the magnitude of the improvement will be very large Small. Because we can see that when the abscissa is at 0.0036, the ordinate has approached 0, and the effect of continuing to increase the temperature on the adsorption performance is almost negligible.
Comment 7: It is quite interesting to observe good regeneration performance of these capsules as adsorbents (Figure 11). Yet, the Authors explained this fact with the “decrease of the mechanical properties of the GO shell” (line 368). What are mechanical properties the Authors 2 mentioned here? It could be unclear for the readers as well as the reasoning that this decrease is responsible for high adsorption amounts.
Response 7: Thinks for your suggestions. In the original text " The main reason for the adsorption amount of the adsorbent may be due to the dispersion of the superhydrophilic GO nanosheets in the aqueous phase, which leads to the decrease of the mechanical properties of the shell layer." This passage explains the adsorption The reason for the decrease of the amount of adsorption to 53% of the first adsorption amount is because of the decrease of the mechanical properties of the shell layer, and the decrease of the mechanical properties brings about a decrease in the adsorption amount rather than an increase.
For mechanical properties, it refers to the mechanical characteristics of materials under different environments (temperature, medium, humidity) under various external loads (tension, compression, bending, torsion, impact, alternating stress, etc.). In this experiment, graphene oxide nanosheets are extremely hydrophilic materials. After several adsorption and desorption processes in water, the mechanical properties of the material decrease, so the adsorption capacity decreases.
Comment 8: The beginning of the conclusion is not clear (what is “this chapter”?). Generally, the conclusion should stress out more the novelty of the work.
Response 8: Thinks for your suggestions. We have revised the beginning of the conclusion, and the revised content will be highlighted in red font.
Comment 9: Generally, the layout of the manuscript and its technical side leave much to be acceptable. For instance, the quality of Figure 12 is clearly worse than others. The citation format should be unified according the guidelines for the authors for this specific journal (brackets in text rather than as upper indexes). The format of equation is also incorrect, the proper punctuation is missing (commas before where, dots after the end of the sentence when an equation is a part of the sentence, especially Eq. (4) and Eq. (12)). The journal requires the addition of details about producers and suppliers (company name, city, country). The Authors used also inconsistent notation for countries (U.S.A. vs. USA). It should be unified.
Response 9: Thinks for your suggestions.
- We have made changes to Figure 12 to make it clear
- We have unified the bibliography format.
- We modified the format of the equation
- We have added details of producers and suppliers (company name, city, country).
- We unified the national symbols.
Comment 10: The language of the manuscript is very clear and understandable, however, the careful proofreading of the whole text is recommended. Some of small issues are listed below: - Perhaps, most common form is: micro- and nano-capsules (line 48). - The notation: 5.0 h looks uncommonly (line 102). - There is a missing space (line 183). - The sentence in line 208 is incomplete. - The sentences in line 245 and 252 should start with the upper case letter. - The number of Eq. (3) is missing (line 268). - The sentence in line 298 is not understandable. - There is an incorrect “in” term in line 383.
Response 10: Thinks for your suggestions. We've revised your question and rechecked the full manuscript and improved formatting errors.

Reviewer 2 Report
The manuscript entitled “Interfacial Imide Polymerization of Functionalized Filled Microcapsule Templates by Pickering Emulsion for Rapid Removal of 3,4,5-Trichlorophenol from Wastewater” presents the possibility of using olive oil filled microcapsules for the removal of chlorophenol contaminants from water.
There are some issues that should be explained and completed before publishing the manuscript.
General remarks:
· The wording “Functionalized Filled Microcapsule Templates” used in the title of the article sounds weird. Moreover, it does not indicate how the templates are functionalized and what they are filled with. The good title of the manuscript should be declarative, engaging and focused.
· In the introductory part the authors wrote that “the massive release of chlorophenol- containing effluents into natural rivers has resulted in serious contamination of surface water, groundwater, and other water bodies”. Do the authors mean any particular region of the world? No details can be derived from the quoted literature. Please refer to the specific reports showing the content of trichlorophenol pollutants in the rivers with an indication of the regions where this problem occurs, e.g. in China (DOI:10.1016/j.chemosphere.2007.10.018).
· Can other vegetable oils (e.g. sunflower oil, which is cheaper than olive oil) also be used to extract water contaminants?
· In the introduction the authors wrote that “common methods for capsule preparation include emulsion interfacial polymerization, interfacial crystallization, and interfacial gelation”. It is worth adding that nowadays microcapsules are also often produced using the electroformation method followed by thermal treatment of Pickering droplets with particle shells (see: doi.org/10.1021/acsami.9b21484 and doi.org/10.1016/j.cap.2022.03.008).
· Are the used materials, i.e. graphene oxide, methylene diisocyanate (HDI) and 1,6-hexanediamine (HMDA) biodegradable and non-toxic? One can find the information that methylene diphenyl diisocyanate is “violently reactive material towards water and other nucleophiles” and hexamethylenediamine “is moderately toxic”. The authors should comment on this.
· In the subsection “2.5. Adsorption isotherm and kinetics experiments” it is written that “the mixture was then shaken … with a contact time of 0-120 min between the sample and the adsorbent”. Why is the upper limit of two hours? What is the long-term stability of microparticles in polluted water?
· How did the authors determine the proportions in which microparticles should be added to the tested solutions? (i.e. 5 mg of microcapsules to 5 mL of the test solutions).
· In the experiment “the mixture was shaken and stirred” and “the adsorbent material was separated from the solution by standing”. How in practice (not in the laboratory) can these microparticles be used for water purification, e.g. in a river?
· Fig. 2a shows that microparticles vary in shape and size. It is also written that “microcapsules, have diverse morphology and uneven size with an average particle size of 90 μm”. What is the particle size distribution? What is the standard error of the mean? Does the shape of the microcapsules affect their adsorption capacity?
· The authors wrote that “the mechanical properties of the microcapsules are poor due to the uneven dispersion of the nanosheets”. Please specify what the word “poor” means in this case? Does it indicate the reduced mechanical strength of the shell? Does this affect their function or stability?
In the manuscript the sentence summarizing this paragraph is a bit surprising “All of the above proved the successful preparation of C-O/W microcapsules”. Does “all” mean: “diverse morphology and uneven size”, “uneven dispersion of the nanosheets” and “poor mechanical properties”)?
· Fig.3d should be described in the text, because the caption of fig. 3 is very laconic.
· Line 215 - “The” in new sentence should be written with a capital letter.
· Since the described microparticles are intended for water purification, what is the purpose of testing their thermal stability at high temperatures (up to 800oC?). Are there any plans to sinter the coatings?
· Adsorption performance studies were carried out at pH = 5.0 (which was experimentally confirmed as the best adsorption condition). It would be good to compare it with measured pH of the trichlorophenol contaminated river water (environmental samples, not the river water, which was artificially polluted in the laboratory by adding chemical compound in certain concentration).
· The long sentence “To further investigate the adsorption characteristics of the adsorbent on the target, this study was performed by adsorption equilibrium experiments in Order to analyze the adsorption equilibrium constant and the maximum adsorption capacity” should be simplified or rephrased.
· Values on the horizontal axis in Figure 9 should be kept separate. The authors can enlarge the image or reduce the number of decimal places.
Author Response
Responses to the Reviewers’ Comments
Dear Prof. Serena Song:
Thank you very much for your letter and for the reviewer’s comments concerning our manuscript entitled “Interfacial Imide Polymerization of Functionalized Filled Microcapsule Templates by Pickering Emulsion for Rapid Removal of 3,4,5-Trichlorophenol from Wastewater” (Manuscript ID: nanomaterials-1924018). We are truly grateful to yours and reviewers’ critical comments and valuable suggestions. Those comments are all valuable and very helpful for revising and improving our paper, as well as the important guiding significance to our researches. Based on these comments and suggestions, we have tried our best to improve the manuscript and made some changes in the revised manuscript in those days. And in the “Response to Decision Letter”, all the changes and also highlighted with red in the revised manuscript and the Supporting Information (corresponding to the serial number in the list). We appreciate for the Editors and Reviewers’ warm work earnestly, and hope that the correction will meet with your approval. Below you will find our point-by-point responses to the reviewers’ comments or questions.
We deeply appreciate your consideration of our manuscript, and if you have any questions, please don’t hesitate to contact us.
Thank you and best regards.
Yours sincerely,
Jianming Pan
Responses to the Reviewer 2’s Comments
Comment 1: In the introductory part the authors wrote that “the massive release of chlorophenol- containing effluents into natural rivers has resulted in serious contamination of surface water, groundwater, and other water bodies”. Do the authors mean any particular region of the world? No details can be derived from the quoted literature. Please refer to the specific reports showing the content of trichlorophenol pollutants in the rivers with an indication of the regions where this problem occurs, e.g. in China (DOI:10.1016/j.chemosphere.2007.10.018).
Response 1: Thinks for your suggestions. We have revised this passage to include the designated regions in the manuscript and cite this document.
Comment 2: Can other vegetable oils (e.g. sunflower oil, which is cheaper than olive oil) also be used to extract water contaminants?
Response 2: Thinks for your suggestions. For the selection of olive oil, we only considered a green solvent with high stability, followed by the separation of chlorophenols through the effect of hydrophilicity and hydrophobicity.
Comment 3: In the introduction the authors wrote that “common methods for capsule preparation include emulsion interfacial polymerization, interfacial crystallization, and interfacial gelation”. It is worth adding that nowadays microcapsules are also often produced using the electroformation method followed by thermal treatment of Pickering droplets with particle shells (see: doi.org/10.1021/acsami.9b21484 and doi.org/10.1016/j.cap.2022.03.008).
Response 3: Thinks for your suggestions. We have incorporated the method you mentioned into the manuscript and the corresponding literature has been cited in the references.
Comment 4: Are the used materials, i.e. graphene oxide, methylene diisocyanate (HDI) and 1,6-hexanediamine (HMDA) biodegradable and non-toxic? One can find the information that methylene diphenyl diisocyanate is “violently reactive material towards water and other nucleophiles” and hexamethylenediamine “is moderately toxic”. The authors should comment on this.
Response 4: Thinks for your suggestions. Graphene oxide is a green material commonly used in the synthesis of adsorbents, while HDI and HMDA are the raw materials for generating microcapsule adsorbents. After the preparation is completed, the remaining HDI and HMDA will be eluted by the solvents and proceeding with reasonable waste liquid.
Comment 5: In the subsection “2.5. Adsorption isotherm and kinetics experiments” it is written that “the mixture was then shaken … with a contact time of 0-120 min between the sample and the adsorbent”. Why is the upper limit of two hours? What is the long-term stability of microparticles in polluted water?
Response 5: Thinks for your suggestions. Because the adsorption equilibrium time of the adsorbent has been known to be about 30 min through adsorption experiments in advance, it is sufficient to control the adsorption time to 120 min, and it is beneficial to the subsequent fitting process.
Because through a series of characterizations, microgels are relatively stable particles, which can maintain good adsorption performance in both acidic and alkaline environments. For wastewater solutions, microcapsules can still maintain high stability.
Comment 6: How did the authors determine the proportions in which microparticles should be added to the tested solutions? (i.e. 5 mg of microcapsules to 5 mL of the test solutions).
Response 6: Thinks for your suggestions. The selection of 5 mg of micro -capsule to 5 ml test solution is because: 1. The output of synthetic micro-capsules is low, so 5 mg is used for adsorption experiments. 2. The concentration of adsorption agent is too high or too low, which will affect the final adsorption effect, so the proportion of 1 mg/ml is used to control the amount of adsorbing solution at 5ml.
Comment 7: In the experiment “the mixture was shaken and stirred” and “the adsorbent material was separated from the solution by standing”. How in practice (not in the laboratory) can these microparticles be used for water purification, e.g. in a river?
Response 7: Thinks for your suggestions. We will take out the water in the river for actual samples (E.G. In A River), and then add the micro -capsule adsorbent agent we prepared to shake it (the purpose is to flow the solution and speed up the adsorption process). Our adsorbent can still achieve the effect of adsorption pollutants, but the adsorption time will be longer.
Comment 8: Fig. 2a shows that microparticles vary in shape and size. It is also written that “microcapsules, have diverse morphology and uneven size with an average particle size of 90 μm”. What is the particle size distribution? What is the standard error of the mean? Does the shape of the microcapsules affect their adsorption capacity?
Response 8: Thinks for your suggestions. The work in this manuscript employs an interfacial imine polymerization method to encapsulate olive oil in a shell composed of graphene oxide and polymer. In terms of characterization, the efficiency of removing contaminants is almost entirely related to the shell thickness and has nothing to do with the size and shape of the capsule, so the particle size distribution and size and shape of the microcapsules are not considered.
Comment 9: The authors wrote that “the mechanical properties of the microcapsules are poor due to the uneven dispersion of the nanosheets”. Please specify what the word “poor” means in this case? Does it indicate the reduced mechanical strength of the shell? Does this affect their function or stability?
In the manuscript the sentence summarizing this paragraph is a bit surprising “All of the above proved the successful preparation of C-O/W microcapsules”. Does “all” mean: “diverse morphology and uneven size”, “uneven dispersion of the nanosheets” and “poor mechanical properties”)?
Response 9: Thinks for your suggestions. Poor means that the mechanical properties of the prepared microcapsules are not good enough and are prone to deformation under large pressure, but even if deformation occurs, the thickness of the shell layer of the microcapsules will hardly change, and its function will not be affected.
The meaning of all means that the fluorescence microscope picture in Figure 2 and the scanning electron microscope picture in Figure 3 both confirm the successful preparation of C-O/W microcapsules.
Comment 10: Fig.3d should be described in the text, because the caption of fig. 3 is very laconic.
Response 10: Thinks for your suggestions. Fig.3 d is a small amount of deformation and rupture observed when the microcapsules were photographed by SEM. For the authenticity of the experiment, it has been explained in the text: “In addition, the mechanical properties of the microcapsules are poor due to the uneven dispersion of the nanosheets.”
Comment 11: Line 215 - “The” in new sentence should be written with a capital letter.
Response 11: Thinks for your suggestions. Thank you again for your suggestion to us. According to the suggestions you mentioned earlier, we have change the sentence in line 215
Comment 12: Since the described microparticles are intended for water purification, what is the purpose of testing their thermal stability at high temperatures (up to 800oC?). Are there any plans to sinter the coatings?
Response 12: Thinks for your suggestions. Thermogravimetric analysis is mainly to prove the successful preparation of C-O/W microcapsules, and it is found that the microcapsules have good thermal stability, so this property is introduced into the article.
However, for the sintered coating, there is no effect on the adsorption properties of the microcapsules in this experiment.
Comment 13: Adsorption performance studies were carried out at pH = 5.0 (which was experimentally confirmed as the best adsorption condition). It would be good to compare it with measured pH of the trichlorophenol contaminated river water (environmental samples, not the river water, which was artificially polluted in the laboratory by adding chemical compound in certain concentration).
Response 13: Thinks for your suggestions. Our laboratory has no conditions to obtain natural trichlorophenol-contaminated river water. If we continue to carry out relevant experiments in the future, we hope to cooperate with you and relevant researchers.
Comment 14: The long sentence “To further investigate the adsorption characteristics of the adsorbent on the target, this study was performed by adsorption equilibrium experiments in Order to analyze the adsorption equilibrium constant and the maximum adsorption capacity” should be simplified or rephrased.
Response 14: Thinks for your suggestions. Thanks again for your suggestion to us. Based on the suggestion you mentioned earlier, we have improved this sentence and highlighted it in red lettering.
Comment 15: Values on the horizontal axis in Figure 9 should be kept separate. The authors can enlarge the image or reduce the number of decimal places.
Response 15: Thinks for your suggestions. Thanks again for your suggestion to us. Following the suggestion you mentioned earlier, we made changes to Figure 9 so that the values on the abscissa remain independent.

Reviewer 3 Report
In this work, olive oil-filled composite capsule (C-O/W) adsorbent was prepared for
adsorption of 3,4,5-Trichlorophenol (3,4,5-TCP) by emulsion templating method. The topic is interesting. A major revision is necessary for reconsideration to publish. The detailed comments on this manuscript are as follows:
My detailed comments are as follows:
1. What is the novelty of this work in comparison to the previous papers? Clarify the novelty from material point of view in the manuscript.
2. Appropriate references must be cited in the result and discussion section of the manuscript e.g. "This change may be related to the degree of ionization of chlorophenols in aqueous solution." suitable references must be added to justify the results obtained.
3. English of the manuscript should be thoroughly checked and corrected.
Author Response
Responses to the Reviewers’ Comments
Dear Prof. Serena Song:
Thank you very much for your letter and for the reviewer’s comments concerning our manuscript entitled “Interfacial Imide Polymerization of Functionalized Filled Microcapsule Templates by Pickering Emulsion for Rapid Removal of 3,4,5-Trichlorophenol from Wastewater” (Manuscript ID: nanomaterials-1924018). We are truly grateful to yours and reviewers’ critical comments and valuable suggestions. Those comments are all valuable and very helpful for revising and improving our paper, as well as the important guiding significance to our researches. Based on these comments and suggestions, we have tried our best to improve the manuscript and made some changes in the revised manuscript in those days. And in the “Response to Decision Letter”, all the changes and also highlighted with red in the revised manuscript and the Supporting Information (corresponding to the serial number in the list). We appreciate for the Editors and Reviewers’ warm work earnestly, and hope that the correction will meet with your approval. Below you will find our point-by-point responses to the reviewers’ comments or questions.
We deeply appreciate your consideration of our manuscript, and if you have any questions, please don’t hesitate to contact us.
Thank you and best regards.
Yours sincerely,
Jianming Pan
Responses to the Reviewer 3’s Comments
In this work, olive oil-filled composite capsule (C-O/W) adsorbent was prepared for adsorption of 3,4,5-Trichlorophenol (3,4,5-TCP) by emulsion templating method. The topic is interesting. A major revision is necessary for reconsideration to publish. The detailed comments on this manuscript are as follows:
Comment 1: What is the novelty of this work in comparison to the previous papers? Clarify the novelty from material point of view in the manuscript.
Response 1: Thinks for your suggestions. There are three main novelties in the article: 1. In this paper, olive oil-filled microcapsules were prepared by the combination of pickering emulsion template method and interfacial in situ imine chemistry method, which is a novel, simple and inexpensive preparation strategy. 2. The olive oil was encapsulated in a shell composed of graphene oxide and polymers, and the target was effectively removed for the adsorption of 3,4,5-TCP through the hydrophobic interaction between olive oil and chlorophenols. 3. The unique microcapsule structure further enhances the kinetic performance, which can reach 92% of the maximum capacity within 40 minutes.
Comment 2: Appropriate references must be cited in the result and discussion section of the manuscript e.g. "This change may be related to the degree of ionization of chlorophenols in aqueous solution." suitable references must be added to justify the results obtained.
Response 2: Thinks for your suggestions. Thanks again for the advice you gave us. Following your previous suggestion, we have added references to the Results and Discussion sections, especially for: "This change may be related to the degree of ionization of chlorophenols in aqueous solution."
Comment 3: English of the manuscript should be thoroughly checked and corrected.
Response 3: Thinks for your suggestions. Thanks again for your suggestion to us. We have checked and corrected all English in the manuscript in accordance with your previously mentioned suggestion, and the revised content has been highlighted in red font.

Round 2
Reviewer 2 Report
Thank you for addressing all my comments.
The manuscript is not the highest quality but probably suitable for the Nanomaterials.